# LLM-STAR: Sequence-Teacher-Anchored LLM Recommender with Adaptive Regularization

## Abstract

Large Language Models (LLMs) have been increasingly adopted for recommendation tasks, yet their ability to leverage the sequential nature of user item interaction data remains underexplored. In this work, we conduct a comprehensive investigation into how LLMs process item sequences and uncover a critical limitation: LLMs often exhibit a set-like prediction behavior, focusing on the unordered collection of items rather than their order. Through experiments where item textual content is removed and only item IDs are provided, we demonstrate that LLMs fail to fully exploit sequential dependencies, leading to degraded sequential recommendation. Motivated by the principle of entropy, we further provide a representation-space perspective: the region occupied by embeddings of ordered item sequences is a compact subspace of that formed by unordered item collections, as sequence information reduces entropy and enforces tighter structure. Building on this insight, we introduce a contrastive learning framework that explicitly guides LLMs to capture sequential patterns by encouraging compact representation of ordered item sequences. Extensive experiments across multiple benchmarks show that our method achieves state-of-the-art performance, surpassing prior LLM-based recommendation approaches.

## 1 Introduction

Since TALLRec (Bao et al., 2023) reformulated sequential recommendation as a language modeling task, an increasing number of approaches have been devoted to pushing the performance boundary of large language models for sequential recommendation (LLM4SR). During this stage, representative works, such as LLaRA (Liao et al., 2024), A-LLMRec (Kim et al., 2024), and CoLLM (Zhang et al., 2025) focused on augmenting LLMs with collaborative signals from traditional models to better exploit their reasoning ability. Recently, researchers(Hou et al., 2024b; Kim et al., 2025) have shifted their attention to the fundamental question of sequential recommendation: whether LLMs can effectively understand the sequential knowledge inherent in the user interaction sequences? Based on the marginal performance gap between LLMs trained on original versus shuffled sequences, Hou et al. (2024b) conclude that LLMs are insensitive to the order of user interactions. Subsequently, Kim et al. (2025) validate this conclusion through extensive experiments and further propose a distillation method that leverage representations from traditional sequential models. In contrast, Zhai et al. (2025) proposes to improve the positional encoding of LLMs, which allows them to be better adapted to sequential recommendation tasks. However, none of these works conducts an in-depth analysis of the underlying reasons why LLMs exhibit insensitivity to sequential order.

In this study, we design a systematic experimental pipeline that encompasses both validation and analysis. Based on our experiments, we draw a more profound conclusion: **in sequential recommendation, the predictive behavior of LLMs is set-like, as they focus on unordered item collections rather than ordered interaction sequences**. Specifically, we begin by validating the observed order-insensitivity of LLMs, with the results summarized in Table 1 and the experimental setup detailed in Section 4.1. We can find that, unlike traditional sequential models, LLM-based models do not rely on the order of user's historical sequence when predicting the next interacted item. Notably, although LLM-SRec (Kim et al., 2025) is designed to mitigate this issue, it continues to exhibit similar behavior on the CDs dataset.

Table 1: Performance of baseline methods trained on original vs. shuffled sequences, evaluated on original sequences (NDCG@10).

| Model | Training | Movies | Scientific | Electronics | CDs |
|---|---|---|---|---|---|
| SASRec | original | 0.3486 | 0.3042 | 0.2474 | 0.3373 |
| | shuffle | 0.2747 | 0.2530 | 0.1828 | 0.3057 |
| | change ratio | (-21.2%) | (-16.8%) | (-26.1%) | (-9.4%) |
| TALLRec | original | 0.1699 | 0.2913 | 0.3098 | 0.3110 |
| | shuffle | 0.1644 | 0.2624 | 0.2394 | 0.3009 |
| | change ratio | (-3.2%) | (-9.9%) | (-22.7%) | (-3.8%) |
| LLaRA | original | 0.3105 | 0.3343 | 0.3017 | 0.3764 |
| | shuffle | 0.2987 | 0.3328 | 0.2936 | 0.3757 |
| | change ratio | (-3.8%) | (-0.4%) | (-2.7%) | (-0.2%) |
| A-LLMRec | original | 0.3376 | 0.3081 | 0.3046 | 0.3622 |
| | shuffle | 0.3208 | 0.3180 | 0.2610 | 0.3584 |
| | change ratio | (-5.0%) | (+3.2%) | (-14.3%) | (-1.0%) |
| LLM-SRec | original | 0.3560 | 0.3388 | 0.3044 | 0.3746 |
| | shuffle | 0.2862 | 0.3066 | 0.2639 | 0.3857 |
| | change ratio | (-19.0%) | (-9.5%) | (-13.3%) | (+3.0%) |

Unlike prior works (Hou et al., 2024b; Kim et al., 2025; Zhai et al., 2025), we further investigate the underlying reasons behind this behavior. To examine whether LLMs rely primarily on textual information or on the sequential relationships among items, we replace the item textual descriptions with their corresponding IDs in the inputs of the LLMs. For this purpose, we focus on TALLRec (Bao et al., 2023) and LLaRA (Liao et al., 2024), as both are finetuned with LoRA (Hu et al., 2022), which facilitates better adaptation to downstream sequential recommendation tasks and strengthens the validity of the observation. As shown in Table 2, once the textual information is removed, LLMs can easily distinguish between the original and shuffled sequences in 75% of the cases. This indicates that, in sequential recommendation, LLMs tend to ignore the order information of user interactions and instead focus on the complete textual content of the entire input sequence. We further analyze the top tokens attended to by LLMs trained separately on the original and shuffled sequences, as shown in Figure 1. In can be observed that the primary attention patterns remain largely unchanged, indicating that as long as the textual information of the input is complete, LLMs can leverage their strong modeling capacity to reconstruct the representation of the interaction sequence.

Based on the above analysis, we can conclude that the representations learned by LLMs for user interaction sequences contain a substantial amount of unordered information. Inspired by the concept of entropy, we tackle this problem from the perspective of the representation space: the representations of ordered item sequences should be a compact subspace formed by unordered item collections. Motivated by this insight, we propose Sequence-Teacher-Anchored LLM Recommender with Adaptive Regularization, i.e., LLM-STAR, which adaptively helps LLMs to learn sequential information through contrastive learning based on the positive and negative anchors in the representation space.

Our contributions can be summarized as follows:

- We reveal the predictive behavior of LLMs in sequential recommendation tasks, demonstrating that they operate in a set-like manner by focusing on collections of items rather than ordered sequences.

- Inspired by the theory of entropy, we interpret the set-like behavior of LLMs from the perspective of the representation space, viewing the representation space of ordered user sequences as a subspace within the representation space of unordered item collections.

Table 2: Results of TALLRec and LLaRA regarding order sensitivity when trained separately with item descriptions and item IDs (NDCG@10).

| Model | Training | Movies | Scientific | Electronics | CDs |
|---|---|---|---|---|---|
| TALLRec | original | 0.1699 | 0.2913 | 0.3098 | 0.3110 |
| | shuffle | 0.1644 | 0.2624 | 0.2394 | 0.3009 |
| | change ratio | (-3.2%) | (-9.9%) | (-22.7%) | (-3.8%) |
| TaLLRec w/o text | original | 0.1611 | 0.1496 | 0.1912 | 0.0950 |
| | shuffle | 0.1582 | 0.1114 | 0.1377 | 0.0859 |
| | change ratio | (-4.8%) | (-25.5%) | (-28.0%) | (-9.6%) |
| LLaRA | original | 0.3105 | 0.3343 | 0.3017 | 0.3764 |
| | shuffle | 0.2987 | 0.3328 | 0.2936 | 0.3757 |
| | change ratio | (-3.8%) | (-0.4%) | (-2.7%) | (-0.2%) |
| LLaRA w/o text | original | 0.2553 | 0.2074 | 0.2722 | 0.1448 |
| | shuffle | 0.2542 | 0.2094 | 0.2295 | 0.1348 |
| | change ratio | (-0.4%) | (+1.0%) | (-15.7%) | (-6.9%) |

| Sequence of Items |
|---|
| [1] "Aretha's Best" [2] "Simply Red - The Greatest Hits" [3] "The Later Years" [4] "Ultimate Grammy Collection: Classic R&B" [5] "Greatest Hits Collection" [6] "Greatest" [7] "Weather" [8] "Greatest Hits" [9] "50 Year Trip: Live at Red Rocks" [10] "Definitive Collection" |

| Layer = 0 | |
|---|---|
| **Original** | **Shuffle** |
| Hits: Hits Grammy Hits Later Collection Greatest Years Classic Collection The | Hits: Hits Grammy Hits Later Collection Greatest Years Classic Collection The |
| Year: Hits Grammy Hits Hits Later Greatest Collection Collection Years Classic | Year: Hits Grammy Hits Hits Later Greatest Collection Collection Years Classic |
| Trip: Year Hits Grammy Hits Hits Later Collection Classic Greatest Collection | Trip: Year Hits Grammy Hits Hits Later Collection Classic Greatest Collection |
| Live: Trip Year Hits Grammy Hits Hits Later Classic Greatest Collection | Live: Trip Year Hits Grammy Hits Hits Later Classic Greatest Collection |
| at: Live Trip Year Hits Grammy Hits Hits Later Classic Collection | at: Live Trip Year Hits Grammy Hits Hits Later Classic Collection |
| Red: at Live Trip Year Hits Grammy Hits Hits Classic Later | Red: at Live Trip Year Hits Grammy Hits Hits Classic Later |
| Rocks: Red at Live Trip Year Hits Grammy Hits Hits Later | Rocks: Red at Live Trip Year Hits Grammy Hits Hits Later |
| Collection: at Hits Trip Live Rocks Year Hits Grammy Hits Red | Collection: at Hits Trip Live Rocks Year Hits Grammy Hits Red |

| Laeyr = -1 | |
|---|---|
| **Original** | **Shuffle** |
| Hits: Collection Red Years Classic Grammy Hits The Later Best Greatest | Hits: Collection Red Grammy Classic Years Best Hits The Later Greatest |
| Collection: Hits Collection Red Classic Years Grammy The Hits Later Best | Collection: Hits Red Collection Classic The Grammy Years Later Best Hits |
| Hits: Collection Collection Hits Red Hits Grammy Classic The Years Best | Hits: Collection Collection Red Hits Grammy The Classic Hits Best Years |
| Year: Grammy Collection Hits Collection Hits Later Years Hits Best Classic | Year: Grammy Collection Later Years Red Hits Best Collection Hits The |
| Trip: Year Collection Collection Grammy Years Hits Hits Later Classic Hits | Trip: Year Grammy Collection Collection Years Red Later Classic Hits Hits |
| Live: Trip Year Grammy Collection Years Collection Hits Hits Classic Red | Live: Trip Year Grammy Years Collection Red Later Classic The Best |
| at: Live Trip Year Grammy Later Classic Years Hits The Best | at: Live Trip Year Grammy Later Classic Years The Collection Red |
| Red: at Live Trip Year Grammy Later Years Hits Collection Red | Red: at Live Trip Year Grammy Later Years Collection Hits Red |
| Rocks: Live Trip at Red Year Collection Collection Hits Years Hits | Rocks: Live Trip at Red Year Collection Hits Collection Grammy Years |
| Collection: Trip Collection Collection Live Grammy Hits Hits Hits Classic Year | Collection: Trip Grammy Collection Live Collection Red Classic Best Hits Hits |

Figure 1: Case study: top tokens attended to by TALLRec trained separately on the original and shuffled sequences for each token in a sample from the CDs dataset. For clarity, only tokens in later positions and those corresponding to complete words are shown.

- To mitigate the order-insensitivity issue of LLMs, we introduce a contrastive learning approach that guides the model to form a compact representation space using positive and negative anchors.

## 2 RELATED WORK

### 2.1 SEQUENTIAL RECOMMENDATION SYSTEMS

Sequential recommendation is a type of personalized recommendation system that captures user interests based on their historical interaction sequences (Ren et al., 2024). In the early stage, Matrix Factorization-based approach (Mnih & Salakhutdinov, 2007; Chaney et al., 2015; He et al., 2017) emerged as the mainstream technology by capturing collaborative signals in the user-item interactions. However, these methods fail to adequately model the evolution of the user interests. Fortunately, the advancement of deep neural networks has effectively mitigated this problem. Representative works include the GRU4Rec (Hidasi et al., 2015) and Caser (Tang & Wang, 2018), which

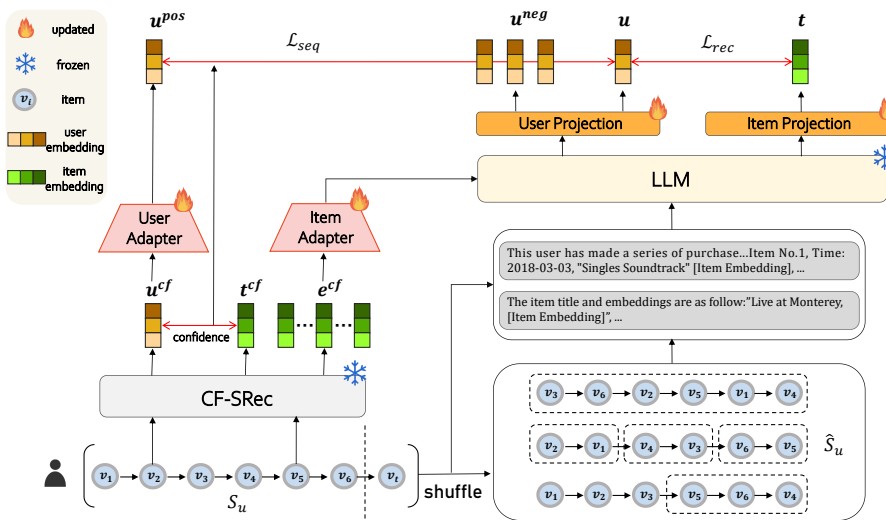

Figure 2: Overall model architecture of LLM-STAR.

leverage the powerful representation ability of the neural networks (Krizhevsky et al., 2012; Cho et al., 2014) to model the user interests. Recent works such as SASRec (Kang & McAuley, 2018) and BERT4Rec (Sun et al., 2019) have further pushed the performance boundaries by introducing attention mechanisms

## 2.2 LLM-BASED SEQUENTIAL RECOMMENDATION SYSTEMS

LLMs are widely applied in sequential recommendation due to their robust real-world knowledge and powerful ability in modeling text sequences (Li et al., 2024; Hou et al., 2024b; Zhang et al., 2025). The pioneering work of TALLRec (Bao et al., 2023) establishes a general paradigm for LLM4SR by formulating interaction sequences as textual prompts fed into LLMs. Building upon this template, subsequent research (Wu et al., 2024; Kong et al., 2024) focuses on incorporating collaborative signals as an independent modality into LLMs to enhance their performance. For instance, LLaRA (Liao et al., 2024) first inserts extra tokens into LLM's prompts, which are converted from item embeddings generated by pre-trained traditional sequential recommenders. Then, the training of LLMs shift from text-only prompts to such hybrid prompts via a curriculum learning approach, which achieves more stable learning. Recent works (Kim et al., 2025; Zhai et al., 2025) start to focus on how to enable LLMs to fully learn the sequential information in the user interactions. LLM-SRec (Kim et al., 2025) validates the insensitivity of LLM-based methods to sequential order, and alleviates this issue by distilling representations from conventional models to guide LLMs in learning sequential knowledge. In contrast, Zhai et al. (2025) recognize that token-level positional encoding in LLMs is not well aligned with sequential recommendation, and establish a new paradigm, which involves introducing an item-order-based positional embedding and an ID generation task.

## 3 METHOD

In this section, we propose LLM-STAR, a simple but effective LLM4SR framework designed to facilitate LLMs to fully recognize sequential information within the interaction sequences. We first present the general LLM4SR loss in Section 3.1, which is also employed in our LLM-STAR, and then introduce the two core modules proposed in our work in the following subsections. Figure 2 illustrates the overall framework of LLM-STAR.

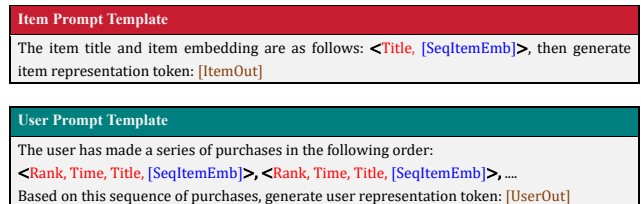

Figure 3: The prompt templates of the candidate item and the user interaction sequence.

## 3.1 LARGE LANGUAGE MODEL FOR SEQUENTIAL RECOMMENDATION

Given the user set $\mathcal{U} = \{u_1, u_2, ..., u_m\}$ and the item set $\mathcal{I} = \{i_1, i_2, ..., i_n\}$, we define the interaction sequence of user $u \in \mathcal{U}$ as $S_u = \{i_1^u, i_2^u, ..., i_{n_u}^u\}$, where the items are ordered chronologically by interaction time. In this work, we set $n_u = 10$, since longer sequences do not provide further benefits(Hou et al., 2024b).

As illustrated in Figure 3, the user's interaction sequence and a candidate item are formulated into a user prompt and an item prompt, respectively. Collaborative information is incorporated via an item adapter $f_{item}$, which maps item embeddings from a pre-trained sequential recommender (Kang & McAuley, 2018) into the semantic space of LLMs. Following the *Next Item Retrieval approach* (Kim et al., 2025), two learnable tokens, [ItemOut] and [UserOut], are inserted into the prompts to obtain the user representation $\mathbf{u^{llm}} \in \mathbb{R}^{d_{llm}}$ and item embedding $\mathbf{i^{llm}} \in \mathbb{R}^{d_{llm}}$.

These representations are then projected into the recommendation space using two projection layers $f_I$ and $f_U$, yielding $\mathbf{u} \in \mathbb{R}^d$ and $\mathbf{i} \in \mathbb{R}^d$. Based on the user historical sequence $S_u$, the interaction intention of user $u$ for the candidate item $i$ can be quantified as $s(\mathbf{u}, \mathbf{i})$, where $s(\cdot)$ denotes the similarity function

For training efficiency, we only consider the last item $t \in S_u$ for each user to construct the sequential recommendation loss, which can be defined as follow:

$$\mathcal{L}_{rec} = -\mathbb{E}_{u \in \mathcal{U}}[\log \frac{e^{s(\mathbf{u},\mathbf{t})}}{e^{s(\mathbf{u},\mathbf{t})} + \sum_{k \in C_u} e^{s(\mathbf{u},\mathbf{k})}}], \tag{1}$$

where $C_u$ is the candidate item set of user $u$.

## 3.2 SEQUENCE-TEACHER-ANCHORED LOSS

To capture sequential dependencies in LLM-based models, we design a sequence-anchored contrastive learning framework, where each sample is paired with a positive instance and multi-scale negative instances in the recommendation space to enforce a compact representation of ordered sequences.

For a given user representation $\mathbf{u}$, we aim to identify an anchor in the representation space that encodes rich sequential information. Shifting $\mathbf{u}$ toward this anchor enable the LLM-based model to better recognize sequential patterns. To this end, we adopt a pre-trained sequential recommender (SASRec) as a teacher, which effectively captures the sequential information in the user interactions (Klenitskiy et al., 2024). For each user interaction sequence $S_u$, the teacher model first produces a user representation $\mathbf{u^{cf}}$ in its own space. A learnable user adapter $f_{user}$ is then employed to transform $\mathbf{u^{cf}}$ into the LLM's recommendation space, yielding the positive anchor representation $\mathbf{u^{pos}} = f_{user}(\mathbf{u^{cf}})$. However, relying solely on the sequential signals provided by the teacher model is insufficient, as LLM-SRec's performance on the CDs dataset demonstrates limited sensitivity to the sequence order, highlighting the necessity of learning from both positive and negative samples.

As shown in Figure 2, we construct three types of negative samples with varying scales and difficulties: (1) random shuffle: the entire sequence is randomly permuted to encourage the LLMs to capture global sequential information; (2) window shuffle: subsequences within windows of varying lengths are shuffled to help the LLMs recognize local sequential patterns; (3) last-$N$ shuffle: the last $N$ items, which are most indicative of the user's immediate next interest (Kang & McAuley, 2018;

Table 3: Statistics of datasets after preprocesing. $|\mathcal{U}|$, $|\mathcal{I}|$, and $|\mathcal{E}|$ denote the number of users, items, and interactions, respectively.

| Dataset | Movies | Scientific | Electronics | CDs |
|---|---|---|---|---|
| $|\mathcal{U}|$ | 12,029 | 23,627 | 27,526 | 18,550 |
| $|\mathcal{I}|$ | 17,672 | 25,764 | 31,778 | 31,202 |
| $|\mathcal{E}|$ | 131,342 | 218,910 | 260,201 | 259,947 |

Qu et al., 2024), are shuffled to strengthen the LLMs' sensitivity to recent behaviors. Finally, these negative samples $\hat{S}_u$ are then fed into the LLMs to generate the multi-scale negative anchors $\mathbf{u^{neg}}$.

Through contrastive learning, pulling the user representation closer to its positive anchor while pushing it away from negative anchors suppresses unordered information, thereby improving the model's capacity to recognize sequential dependencies. The optimization objective can be defined as follows:

$$\mathcal{L}_{seq} = -\mathbb{E}_{u \in \mathcal{U}}[\log \frac{e^{s(\mathbf{u},\mathbf{u^{pos}})}}{e^{s(\mathbf{u},\mathbf{u^{pos}})} + \sum_{i \in \{1,...,|\hat{S}_u|\}} e^{s(\mathbf{u},\mathbf{u_i^{neg}})}}]. \tag{2}$$

### 3.3 ADAPTIVE REGULARIZATION

Although the contrastive learning mechanism proposed in Section 3.2 can theoretically help the model learn a compact representation subspace, there still exist some potential issues. In particular, the teacher model provides strong sequential knowledge, yet its reliability is not always guaranteed (Cui et al., 2024). Besides, some samples inherently exhibit weak sequential patterns (Kang & McAuley, 2018; Klenitskiy et al., 2024), forcing the model to learn sequential knowledge from the original and shuffled samples in such cases may lead to overfitting to noisy orders. Then, the similarity between them, denoted as the confidence $c_u = s(\mathbf{u^{cf}}, \mathbf{t^{cf}})$, quantifies the teacher's certainty about the sequential knowledge in $u$'s interaction sequence. The higher the confidence, the stronger the sequential information of the sample. We utilize this confidence as the adaptive weight for each sample and rewrite Eq.2 as follows:

$$\mathcal{L}_{seq} = -\mathbb{E}_{u \in \mathcal{U}}[s(\mathbf{u^{cf}}, \mathbf{t^{cf}}) \log \frac{e^{s(\mathbf{u},\mathbf{u^{pos}})}}{e^{s(\mathbf{u},\mathbf{u^{pos}})} + \sum_{i \in \{1,...,|\hat{S}_u|\}} e^{s(\mathbf{u},\mathbf{u_i^{neg}})}}]. \tag{3}$$

The final optimization objective of LLM-STAR can be described as follows:

$$\mathcal{L}_{rec} + \beta \mathcal{L}_{seq}, \tag{4}$$

where $\beta$ denotes the hyper-parameter used to scale the auxiliary loss.

## 4 EXPERIMENT

In this section, we conduct comprehensive experiments to answer the following research questions (**RQs**) and thus demonstrate the effectiveness of LLM-STAR.

- **RQ1**: How does LLM-STAR perform compared to existing LLM-based sequential recommendation models on real-world industrial datasets?
- **RQ2**: Whether the proposed Sequence-Teacher-Anchored loss and the Adaptive Regularization modules work?
- **RQ3**: How does the hyper-parameter $\beta$ affect the performance of LLM-STAR?
- **RQ4**: Whether the sequential knowledge learned by LLM-STAR can generalize to other domains?

### 4.1 EXPERIMENT SETUP

**Datasets.** All baselines and our LLM-STAR are evaluated on four Amazon benchmarks (Hou et al., 2024a): Movies, Scientific, Electronics, and CDs. Following the same preprocessing as Kim et al.

Table 4: Overall performances of the baseline models and LLM-STAR. The best performances are denoted in bold.

| Dataset | Metric | SASRec | TALLRec | LLaRA | A-LLMRec | LLM-SRec | LLM-STAR |
|---------|--------|--------|---------|-------|----------|----------|----------|
| Moveis | NDCG@10 | 0.3486 | 0.1699 | 0.3105 | 0.3376 | 0.3560 | **0.3630** |
| | HR@10 | 0.5268 | 0.3256 | 0.5038 | 0.5313 | 0.5569 | **0.5596** |
| Scientific | NDCG@10 | 0.3042 | 0.2913 | 0.3343 | 0.3081 | 0.3388 | **0.3511** |
| | HR@10 | 0.4957 | 0.4926 | 0.5464 | 0.5143 | 0.5532 | **0.5708** |
| Electronics | NDCG@10 | 0.2474 | **0.3098** | 0.3017 | 0.3046 | 0.3044 | 0.3040 |
| | HR@10 | 0.4121 | 0.4933 | 0.4878 | **0.4965** | 0.4885 | 0.4905 |
| CDs | NDCG@10 | 0.3373 | 0.3111 | 0.3764 | 0.3622 | 0.3746 | **0.3852** |
| | HR@10 | 0.5041 | 0.5092 | 0.6054 | 0.5981 | 0.5986 | **0.6060** |

(2025), we also filter out users and items with fewer than five interactions, as cold start is beyond the scope of our work. The specific statistics of each dataset are summarized in Table 3.

**Baselines.** We compare LLM-STAR with the traditional sequential recommendation method SAS-Rec (Kang & McAuley, 2018) and four LLM-based methods, i.e., TALLRec (Bao et al., 2023), LLaRA (Liao et al., 2024), A-LLMRec (Kim et al., 2024), and LLM-SRec (Kim et al., 2025). Considering that larger LLM sizes do not necessarily yield better performance while incurring higher inference latency and resource overhead (Qu et al., 2024; Xu et al., 2024), we adopt LLaMA 3.2 (3B-Instruct) as the backbone for all LLM4SR baselines.

**Evaluation Protocol.** We adopt the widely used leave-last-out evaluation protocol (Liu et al., 2024; Sun et al., 2025; Liu et al., 2025). Concretely, for each interaction sequence, the last two items are held out for validation and test, respectively, while the remaining items are used for training. During evaluation, we pair the positive item with 99 negative samples to form a candidate set, and report the results of Hit Ratio (HR@10) and Normalized Discounted Cumulative Gain (NDCG@10). When verifying the order-insensitivity of LLMs, we adopt the same shuffling strategy as Kim et al. (2025). The item descriptions together with their embeddings in the user interaction sequences are shuffled only once before training, while preserving timestamps unchanged to prevent information leakage.

**Implement Details.** The output dimension of the teacher model is set to 64. The MLP layers, i.e., $f_I$, $f_U$, $f_{item}$, and $f_{user}$ are configured with a hidden size of 2048 and an output dimension of 128. We use Adam optimizer (Kingma & Ba, 2014) with a learning rate of 0.0001. The batch size is set to 20 for all datasets, except for Electronics, where it is 16. The hyper-parameter $\beta$ is set to 0.4 for Movies, Scientific, and Electronics, and 0.7 for CDs. Models are trained for 10 epochs, with 10% of each epoch reserved for validation and an eartly stopping patience of 10. All experiments are conducted on a single NVIDIA L40 GPU (45GB) and a single NVIDIA A800 GPU (80GB).

## 4.2 PERFORMANCE COMPARISONS (**RQ1**)

Table 4 presents the performance comparison between LLM-STAR and the baseline methods on four industrial datasets, from which we have the following observations.

❶ *Except on Electronics, SASRec achieves comparable and even better performance than several LLM-based baselines.* Although LLaRA and A-LLMRec leverage the item embeddings or user representations from the pre-trained SASRec, they still fail to surpass SASRec in the overall performance. Moreover, both these LLM-based methods are substantially more complex, indicating that their capacity to capture sequential information is not yet fully exploited.

❷ *LLM-STAR significantly outperforms the teacher model and achieves the best overall performance.* Through the adaptive weighting strategy to the sequence-teacher-anchored loss, LLM-STAR effectively balances the knowledge distilled from the teacher, achieving superior performance over it. Moreover, the design of positive and negative anchors enables LLMs to capture sequential information that was previously overlooked, thereby fully leveraging their powerful reasoning capability.

Table 5: NDCG@10 scores for the ablation study.

| Version | Ablation | Movies | Scientific | Electronics | CDs |
|---------|----------|--------|------------|-------------|-----|
| (a) | w.o. STA & AR | 0.3204 | 0.3088 | 0.2659 | 0.3619 |
| (b) | w.o. AR | 0.3491 | 0.3405 | 0.2855 | 0.3785 |
| (c) | LLM-STAR | **0.3630** | **0.3506** | **0.3040** | **0.3852** |

❸ *LLM-STAR outperforms LLM-SRec, another state-of-the-art approach that aims to enhance the ability of LLM-based models to capture sequential information.* By roughly distilling SASRec's user representations as teacher signals, LLM-SRec fails to recognize the intrinsic prediction patterns of LLM-based models. This leads to excessive dependence on the teacher's sequential knowledge and compromises the model's capacity to distinguish negative samples. In contrast, guided by the representation-space perspective, we naturally introduce multi-scale negative anchors, achieving superior performance.

❹ *LLM-STAR achieves the suboptimal performance on Electronics.* As shown in Table 1, the baseline methods experience the largest performance drop on the shuffled Electronics dataset, indicating that it contains the richest sequential information among the four benchmarks. This allows the models to directly capture sufficient sequential information, which also explains why LLM-based models exhibit comparable performance on this dataset.

### 4.3 ABLATION STUDY (**RQ2**)

To examine the contribution of each module in LLM-STAR to the model's performance, we perform the ablation study. As shown in Table 5, we gradually remove the two key modules, **A**daptive **R**egularization and **S**equence-**T**eacher-**A**nchored loss, from LLM-STAR to investigate their effectiveness. We can make the following observations.

❶ *Compared with the vanilla version (a), incorporating sequence-teacher-anchored loss significantly enhances the model's performance.* Across the four datasets, version (b) achieves an average 7.6% improvement over version (a). This demonstrates that setting anchors in the representation space effectively helps LLMs learn sequential information and providing empirical support for our proposed perspective.

❷ *The introduction of adaptive regularization further pushes the model beyond its performance bottleneck.* By weighting the sequence-teacher-anchored loss according to the teacher model's confidence in the sequential information of each sample, the model learns more cautiously when the teacher signals are unreliable or the sample contains limited sequential patterns, thereby achieving the enhanced performance.

### 4.4 HYPER-PARAMETER ANALYSIS (**RQ3**)

We analyze the impact of the hyper-parameter $\beta$ on the model performance, since it determines the contribution of our proposed loss item. In particular, we set $\beta$ in the range of 0.3 to 0.9 and present the results of LLM-STAR on CDs, as shown in Figure 4. We can draw the following conclusions.

❶ *Introducing a suitable level of sequential information can enhance model's performance.* A non-monotonic trend in both NDCG@10 and HT@10 with respect to $\beta$ is observed, with values initially increasing and then declining. This indicates that moderately increasing $\beta$ can effectively enhance LLM's ability to learn sequential information, whereas an excessively large $\beta$ will cause the model to overlook fitting the recommendation task, resulting in degraded performance.

❷ *LLM-STAR is insensitive to the hyper-parameter $\beta$.* LLM-STAR maintains consistently strong performance for $\beta$ between 0.5 and 0.7, demonstrating the robustness of our approach.

### 4.5 PERFORMANCE UNDER CROSS DOMAIN SCENARIO(**RQ4**)

We conduct cross-domain experiments to assess whether sequential knowledge learned by LLM-STAR can be transfer from the source domain to an unseen target domain. The model is pre-trained

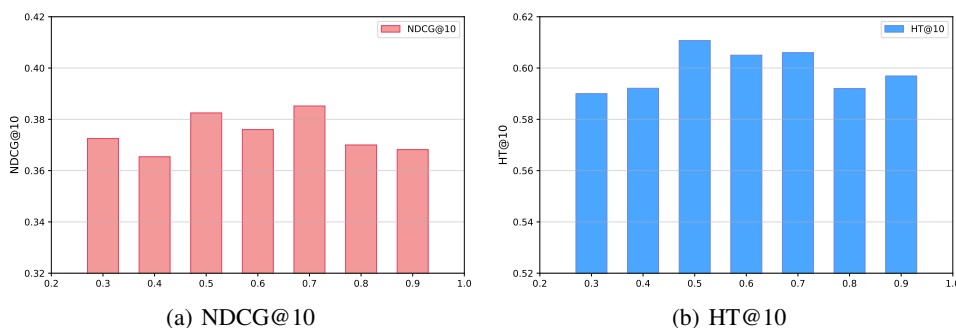

(a) NDCG@10           (b) HT@10

Figure 4: Experimental results of LLM-STAR regarding the hyper-parameter $\beta$ on CDs.

Table 6: Performance of the cross domain experiments. To better compare the performance differences between traditional LLM-based methods and those focus on sequential information, we only consider TALLRec, LLaRA, LLM-SRec, and LLM-STAR.

| Cross Domain | Metric | TALLRec | LLaRA | LLM-SRec | LLM-STAR |
|---|---|---|---|---|---|
| Scientific → CDs | HT@10 | 0.1276 | 0.0946 | 0.1619 | **0.1659** |
| | NDCG@10 | 0.0606 | 0.0422 | 0.0763 | **0.0782** |
| Scientific → Movies | HT@10 | 0.1108 | 0.0887 | 0.2617 | **0.3025** |
| | NDCG@10 | 0.0478 | 0.0382 | 0.1340 | **0.1495** |
| CDs → Scientific | HT@10 | 0.0970 | **0.1749** | 0.1168 | 0.1666 |
| | NDCG@10 | 0.0436 | 0.0837 | 0.0529 | **0.0945** |
| Moviess → Scientific | HT@10 | 0.1164 | 0.1561 | 0.2006 | **0.2185** |
| | NDCG@10 | 0.0524 | 0.0751 | 0.0929 | **0.0988** |

on Scientific as it provides sufficient data and contains more implicit sequential information compared to Electronics, and evaluated on CDs and Movies. Reverse transfer experiments are also performed for comprehensive validation. As shown in Table 6, we can make the following conclusion.

❶ *LLM-STAR can effectively transfer the learned sequential knowledge.* In both transfer directions, LLM-STAR achieves the best performance, indicating that our model can effectively capture sequential information and retain its advantages when transferred to unseen domains.

❷ *Enabling the model to learn sequential knowledge from multi-scale negative samples is beneficial.* LLM-SRec's performance advantage decreases when transferring from Movies and CDs to Scientific, as the source datasets exhibit less pronounced sequential patterns. This indicates that relying solely on positive samples from the teacher model is insufficient.

## 5 CONCLUSION

In this work, we reveal the set-like prediction behavior of LLMs in sequential recommendation, where the next-item prediction depends on the collection of items rather than their orders. Inspired by the concept of entropy, we provide a representation-space perspective, the space occupied by the representations of ordered item sequence is a subspace of that formed by the unordered item collections. Based on this insight, we propose LLM-STAR, a contrastive learning framework that encourages LLM-based models to learn compact representation spaces by generating the positive and multi-scale negative anchors. Experiments on four industrial datasets demonstrate that our method achieves state-of-the-art performance.

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
