# OpenReview forum: "LLM-STAR: Sequence-Teacher-Anchored LLM Recommender with Adaptive Regularization"
_ICLR.cc/2026/Conference — ICLR 2026 Conference Withdrawn Submission_

### Official Review · Reviewer_fwnq · 2025-10-26

**Soundness:** 3
**Presentation:** 3
**Contribution:** 2
**Rating:** 2
**Confidence:** 4

**Summary:**

The paper investigates why LLM-based sequential recommenders often appear weakly sensitive to interaction order, diagnosing a set-like treatment of histories on Amazon Reviews via targeted probes (e.g., shuffling and ID-only inputs). It proposes LLM-STAR—a lightweight, sequence-teacher–anchored and adaptively weighted regularization that plugs into standard retrieval training—to encourage order-aware representations, with analyses and ablations indicating practical improvements without architectural changes.

**Strengths:**

1. **Clear and useful conceptual diagnosis of LLM behavior in sequential recommendation.** The paper articulates a coherent perspective that LLMs tend to treat user histories in a set-like manner, and it frames “order-aware” representations as a compact subspace within the broader unordered semantic space; this organizing view, supported by targeted probes (e.g., shuffling analyses, attention inspections, and ID-only inputs), offers a productive lens for reasoning about when and why order may or may not emerge in LLM-based recommenders without overclaiming beyond the examined setting.

2. **Methodologically novel yet lightweight regularization via a sequence teacher.** The sequence-teacher–anchored objective steers LLM user representations toward order-aware anchors while pushing against carefully designed order-corrupted negatives, and the adaptive weighting mitigates the risk of over-enforcing noisy sequential cues; importantly, the loss integrates seamlessly with standard retrieval training, making the approach modular, orthogonal to backbone choices, and easy to adopt in existing LLM-for-recs pipelines.

3. **Practical training recipe with informative internal analyses.** The approach is compatible with parameter-efficient fine-tuning and does not require architectural changes or specialized pretraining, which enhances practical deployability; moreover, the accompanying ablation-style studies and probing experiments are thoughtfully designed to isolate the contribution of each component within the proposed framework, and preliminary cross-domain observations suggest the method has potential to transfer sequential inductive bias without asserting claims that depend on strongly order-centric benchmarks.

**Weaknesses:**

1. **Baselines against modern sequential recommenders are insufficient.** The paper compares primarily to a single non-LLM method from 2018, which is not representative of the substantial progress made in sequential recommendation over the last several years; a fair assessment requires methodological discussion and empirical comparisons to more recent non-LLM sequential models (e.g., stronger transformer-based and contrastive variants), along with updated training protocols and evaluation practices so that the claimed advantages are credible and contextually grounded.

2. **Dataset choice weakly reflects order-dependent behavior.** The Amazon Reviews setting does not emphasize fine-grained sequence dynamics to the same degree as short-video or music recommendation, where immediate temporal context and session-level dependencies are stronger; consequently, it is unsurprising that a model appears less sensitive to order on this dataset, and performance there may correlate only weakly with true sequence-modeling capability, suggesting that additional experiments on datasets with clearer temporal signals would better validate the central claims.

3. **Shuffling-based evidence does not substantiate that order sensitivity is beneficial.** Table 1 is interpreted as showing that LLMs are not particularly sensitive to order, yet language order often has limited impact on overall meaning and robustness to small permutations is a known property; moreover, observing a larger degradation under shuffled inputs does not in itself imply a better model, as it can equally indicate reduced robustness to the specific transformation, and the paper does not persuasively justify why order should be expected to matter for this dataset, leaving the causal link between order information and recommendation quality underexplored.

4. **LLMs should be capable of learning order; observed insensitivity likely stems from limited fine-tuning.** Given strong positional encodings and modeling capacity (e.g., rotary position embeddings), an end-to-end optimization regime would reasonably allow an LLM to capture sequential dependencies at least as well as dedicated recommenders; the reported phenomenon more plausibly reflects the partial adaptation of the LLM—where fine-tuning preserves its robustness to natural-language order variations—so the behavior highlighted is arguably an advantage rather than a deficiency, and a deeper analysis contrasting full end-to-end training with the current adaptation setup would clarify this point.

**Questions:**

see weaknesses.

---

### Official Review · Reviewer_heJn · 2025-10-30

**Soundness:** 3
**Presentation:** 3
**Contribution:** 1
**Rating:** 2
**Confidence:** 4

**Summary:**

This paper proposes LLM-STAR, a sequentially aware LLM-based recommender model that reinforces sequential knowledge from a representation-space perspective. The approach is motivated by the observation that existing LLM-based recommenders tend to ignore item order and conduct set-like behavior. Extensive experiments on multiple Amazon datasets—including ablation analyses, baseline comparisons, and hyperparameter sensitivity tests—verify the effectiveness of the proposed method.

**Strengths:**

1. The authors tackle a key limitation of existing LLM-based recommender systems — their inability to effectively capture the sequential order of user–item interactions.
2. The authors introduce a novel approach, LLM-STAR, which enables LLM-based models to become sequence-aware by incorporating sequential information through a representation-space framework.
3. Experimental results show that LLM-STAR achieves substantial performance gains over prior LLM-based recommender methods.

**Weaknesses:**

1. The observation regarding the set-like behavior of LLM-based recommender models is not novel. Prior work [1] has already demonstrated that these models struggle to capture the sequential order of items, which reflects the same phenomenon. Thus, the term “set-like behavior” appears to be a rephrasing of an existing finding.
* Additionally, the analysis associated with Table 2 and Figure 1 requires further clarification. For Table 2, to substantiate the claim that LLMs primarily rely on complete text content rather than order, the authors should include an additional experiment using only textual descriptions (without item IDs) and compare performance differences between original and shuffled sequences. For Figure 1, the purpose and interpretation are ambiguous—the paper does not clearly explain what the experiments on different layers indicate, what the red-highlighted text signifies, or what top tokens attended by the LLM are.

2. The motivation for incorporating sequence awareness in ordered item sets is not clearly addressed. The authors simply shows performance improvements but do not provide deeper analytical evidence showing that the proposed model truly captures or understands item order.

3. Further analysis is needed to clarify whether LLM-STAR indeed learns sequential information. Since the model uses both item embeddings and text descriptions while keeping the LLM frozen, it remains uncertain whether performance gains stem from the LLM’s comprehension of sequence or from external representations. Additional ablation studies—such as removing item embeddings or text descriptions—should be performed to explicitly examine how the LLM interprets sequential user interactions.

4. Although adherence to the Code of Ethics is required, the paper does not include an explicit statement confirming compliance. This omission raises concerns about whether the authors have fully met ethical publication requirements.

[1] Lost in Sequence: Do Large Language Models Understand Sequential Recommendation? Kim et al., KDD'25

**Questions:**

Question.

1. Why is sequential knowledge beneficial for cross-domain recommendation? Please provide specific reasoning or analysis explaining the source of the observed performance improvements.

2. Why do the dataset statistics differ from those reported in the previous work [1], despite the claim in Line 323 that this study follows the same data preprocessing procedure?

3. Could the authors clarify the main differences between this work and the previous study [1]? It appears that the proposed method heavily relies on the earlier findings and offers the simple method

Given that the primary finding is not new and the proposed approach is relatively simple, the paper’s overall contribution to the LLM-based recommender systems community seems marginal. Therefore, I recommend rejection.

[1] Lost in Sequence: Do Large Language Models Understand Sequential Recommendation? Kim et al., KDD'25

---

### Official Review · Reviewer_nHEf · 2025-10-30

**Soundness:** 2
**Presentation:** 1
**Contribution:** 2
**Rating:** 2
**Confidence:** 4

**Summary:**

This paper explores the challenge of large language models (LLMs) being insensitive to sequence information in sequence recommendation tasks. It proposes a contrastive learning-based approach to improve LLMs' ability to capture and model sequence information, supported by extensive experimental validation. However, the paper has notable shortcomings in writing quality, novelty articulation, technical contributions, and clarity of argumentation, making it unsuitable for acceptance at ICLR 2026.

**Strengths:**

- The paper offers an interesting perspective, suggesting that the predictive behavior of LLMs is ensemble-oriented, with a stronger focus on unordered sets of items.

- It presents a comprehensive survey of sequence recommendation methods, covering both traditional approaches and LLM-based techniques.

**Weaknesses:**

- Writing quality: The paper lacks clarity, making it challenging to discern key insights and the research motivation.
- Limited novelty: The proposed approach is not innovative, as adapter-based representation mapping with contrastive learning has been widely explored in prior studies for guiding LLMs in capturing sequential and cooperative information.
- Empirical limitations: The explanation in Table 1 regarding LLMs' insensitivity to observed order is unclear and insufficiently justified.
- Reproducibility: The absence of provided code raises concerns about reproducibility.

**Questions:**

- What are the specific settings for the "original" and "shuffle" options in Table 1? Section 4.1 does not clearly explain whether these settings are based on a traditional baseline or a LLM-based baseline. Additionally, the empirical results in Table 1 lack detailed discussion—how do they explain the insensitivity of LLMs to observed sequences?

- The paper claims that existing work has not deeply analyzed the reasons for LLMs' insensitivity to sequence order. Where is the convincing reasoning or analysis provided in this paper?

- What advantages do LLMs offer in understanding sequence knowledge compared to traditional sequence modeling methods?

- Why does the proposed method rely on contrastive learning, as seen in mainstream LLM-enhanced recommendation methods? Can your framework effectively enable LLMs to understand the order information in interactive sequences? What are the key differences between your approach and others?

---

### Official Review · Reviewer_pjMy · 2025-11-01

**Soundness:** 2
**Presentation:** 2
**Contribution:** 2
**Rating:** 2
**Confidence:** 3

**Summary:**

This paper addresses a critical limitation of Large Language Models (LLMs) in sequential recommendation: LLMs exhibit set-like prediction behavior, focusing on unordered item collections rather than the order of user interactions. Through systematic experiments (e.g., removing item textual content and analyzing attention patterns), the authors verify this order insensitivity and provide a representation-space perspective inspired by entropy—ordered sequence embeddings form a compact subspace of unordered collection embeddings. To mitigate this issue, they propose LLM-STAR, a framework integrating two core modules: (1) Sequence-Teacher-Anchored (STA) loss, which uses a pre-trained sequential model (SASRec) as a teacher to generate positive anchors and multi-scale negative anchors (random shuffle, window shuffle, last-N shuffle) for contrastive learning; (2) Adaptive Regularization (AR), which weights the STA loss by the teacher’s confidence in sequential patterns to avoid overfitting to noisy data. Extensive experiments on four Amazon datasets demonstrate that LLM-STAR achieves state-of-the-art performance, outperforming baseline LLM-based and traditional sequential recommendation models.

**Strengths:**

## 1. Originality.
- The paper reveals the set-like prediction behavior of LLMs in sequential recommendation through rigorous experiments, which is a deeper and more fundamental insight than prior works (e.g., Hou et al., 2024b; Kim et al., 2025) that only observed order insensitivity without exploring the underlying mechanism.
- The entropy-inspired representation-space perspective provides a novel theoretical explanation for LLM’s order insensitivity, linking sequential information to embedding compactness.
- mThe combination of teacher-guided contrastive learning with multi-scale negative anchors and adaptive regularization is a creative design, addressing the limitations of prior distillation-based methods (e.g., LLM-SRec) that over-rely on positive teacher signals.

**Weaknesses:**

## 1. Limited analysis of textual information’s role
The paper shows that removing textual content increases LLM’s order sensitivity (Table 2), but it does not further explore why textual information dominates LLM’s prediction. In fact, I believe that the text dominance may be due to the author using LLM as the base for training, with the amount of training data being far less than that of Pretrain. A data volume at the tens of millions level or even the industrial level might directly solve this problem, but the author did not make a comparison in this regard. Therefore, this directly leads to my doubt about the author's motivation and their proposition.


## 2. Lack of efficiency analysis.
- Given that LLM-STAR introduces additional modules, it is critical to demonstrate that the performance gain is not at the cost of excessive computational overhead, especially for industrial applications.

## 3. Limited exploration of negative anchor design.
- The paper uses three types of negative anchors but does not analyze the contribution of each type individually (e.g., which shuffle strategy is most effective for capturing sequential patterns).
- There is no comparison with other negative sampling strategies (e.g., hard negative sampling based on item similarity) to verify if multi-scale shuffle-based negatives are optimal.

**Questions:**

- Conduct an ablation study on individual negative anchor types to clarify their respective contributions and optimize the negative sampling strategy.
- Add a case study to visualize how LLM-STAR’s embeddings of ordered sequences become more compact compared to baseline LLMs, verifying the representation-space perspective.
- Conduct additional validation experiments with an extremely large data scale and re-explore the proposed set-like prediction behavior based on these experiments.

**Details Of Ethics Concerns:**

No concern

---

### Note · Authors · 2025-11-14

I have read and agree with the venue's withdrawal policy on behalf of myself and my co-authors.